# Spatial and Temporal Distribution Characteristics and Potential Risks of Sulfonamides in the Shaanxi Section of the Weihe River

**DOI:** 10.3390/ijerph19148607

**Published:** 2022-07-15

**Authors:** Lei Duan, Siyue Yang, Yaqiao Sun, Fei Ye, Jie Jiang, Xiaomei Kou, Fan Yang

**Affiliations:** 1School of Water and Environment, Chang’an University, Xi’an 710054, China; duanlei1978@126.com (L.D.); ysy19981005@163.com (S.Y.); yefei6668@163.com (F.Y.); 13368335916@163.com (J.J.); 2Key Laboratory of Subsurface Hydrology and Ecological Effects in Arid Region, Ministry of Education, Chang’an University, Xi’an 710054, China; 3Shaanxi Union Research Center of University and Enterprise for River and Lake Ecosystems Protection and Restoration, Xi’an 710065, China; kouxm0427@163.com (X.K.); paper_yangfan@163.com (F.Y.); 4Power China Northwest Engineering Corporation Limited, Xi’an 710065, China

**Keywords:** emerging contaminants, sulfonamides, Shaanxi section of the Weihe River, ecological risk assessment, health risk assessment

## Abstract

The hazards of antibiotics as emerging contaminants to aquatic ecosystems and human health have received global attention. This study investigates the presence, concentration levels, spatial and temporal distribution patterns, and their potential risks to aquatic organisms and human health of sulfonamides (SAs) in the Shaanxi section of the Weihe River. The SA pollution in the Weihe River was relatively less than that in other rivers in China and abroad. The spatial and temporal distribution showed that the total concentrations of SAs in the Weihe River were highest in the main stream (ND–35.296 ng/L), followed by the south tributary (3.718–34.354 ng/L) and north tributary (5.476–9.302 ng/L) during the wet water period. Similarly, the order of concentration from highest to lowest during the flat water period was main stream (ND–3 ng/L), north tributary (ND–2.095 ng/L), and south tributary (ND–1.3 ng/L). In addition, the ecological risk assessment showed that the SAs other than sulfadiazine (SDZ) and sulfamethoxazole (SMZ) posed no significant risk (RQS < 0.01) to the corresponding sensitive species during both periods, with no significant risk to human health for different age groups, as suggested by the health risk assessment. The risk of the six SAs to both aquatic organisms and human health decreased significantly from 2016 to 2021.

## 1. Introduction

Antibiotics are a group of antibacterial drugs produced by bacteria and fungi or synthesized artificially and limit bacterial growth [1]. These are widely utilized in animal husbandry, aquaculture, and medicine. China is the largest producer and consumer of antibiotics globally, with the overall usage of antibiotics reaching 162,000 tons in 2015, accounting for almost 50% of the worldwide consumption [2]. Four major groups of emerging contaminants are now receiving global attention: persistent organic pollutants, endocrine-disrupting chemicals, antibiotics, and microplastics [3,4]. Antibiotic contamination may occur in various ways, including percolation into groundwater, soil contamination through irrigation, and even entry into aquatic organisms, all of which can impact human health via the food chain. Prolonged exposure to antibiotics may cause drug resistance, accelerate the evolution and spread of antibiotic resistance genes (ARGs), and contribute to the emergence of drug-resistant strains, posing a potential threat to human health and ecosystems [5]. Several basins in China currently contain antibiotics, primarily sulfonamides, quinolones, tetracyclines, and macrocyclic lipids [6]. The contamination caused by antibiotics is primarily the result of improper treatment of wastewater generated by pharmaceutical companies, sewage treatment plants, households, livestock, and agriculture [7]. In contrast, traditional wastewater treatment processes are unable to completely remove antibiotics from water, resulting in the transfer of large quantities of antibiotics into surface water, groundwater, and even drinking water. Most antibiotics cannot be completely metabolized after entering the organism, and 50–90% of them ultimately reach the aquatic environment through urine or feces [8]. Antibiotic concentrations in China have reached nanogram or even milligram levels today [9], as they are difficult to degrade and easy to enrich in the ecosystem due to their environmental persistence and bioaccumulation. Thus, they may accumulate in the environment and organisms for an extended period of time. Consequently, it has become a new kind of water pollution.

The fundamental chemical structure of sulfonamides, which is p-Aminobenzenesulfonamide [10], aids in achieving bacteriostatic and sterilizing effects [11] by inhibiting the synthesis of folic acid and bacterial nucleoproteins. The major sulfonamides in the aqueous environment include sulfadiazine (SDZ), sulfathiazole (STZ), sulfapyridine (SPZ), sulfamethazine (SM2), sulfamonomethoxine (SMM), sulfachloropyridazine (SCP), sulfamethoxazole (SMZ), sulfamethizole (SMI), sulfamethoxypyrimidine (SMR), sulfadimethoxine (SDM), sulfamethoxypyridazine (SMP), sulfaquinoxaline (SQZ), etc. There have been numerous studies on antibiotics confirming the presence of sulfonamides in Chinese rivers such as the Pearl River basin [12], the Yangtze River basin [13], the Haihe River basin [14], and the Liaohe River basin [15], as well as in several foreign water environments. The concentrations of SDZ and SPZ in the surface water of Lower Saxony in Germany reached 147 ng/L and 155 ng/L, respectively, while that of SMZ reached up to 1140 ng/L [16]. The highest concentration of SMZ of 4330 ng/L was reported in the urban canals in the northern region of Vietnam [17]. However, the concentration of SMZ was comparably higher in urban canals in Hanoi than in North America, Europe, and other Asian countries [18].

The Yellow River, the second largest river in China, is an important source of water for industrial, agricultural, and domestic use in northern China. Currently, the Yellow River basin has been plagued by varying degrees of antibiotic contamination, with the pollution level in tributaries often more severe than that in the main stream. The Weihe River, the largest tributary of the Yellow River, which is located in the center of the Yellow River basin, is the “heartland” of Chinese civilization, with a basin area about 1/10 of that of the Yellow River [19]. Current research on sulfonamides pollution in the Weihe River has mainly focused on a particular river section without considering the entire watershed and has only evaluated the ecological risk to aquatic biota without evaluating the risk to human health. Although the water quality of the Weihe River has significantly improved since 2018 [20], the evolution of antibiotic residues remains uncertain. In this context, the Shaanxi section of the Weihe River is selected as the study area for the analysis of the types of antibiotics and their concentration levels in main stream and tributaries, understanding of the characteristics of spatial and temporal distribution, and the assessment of the potential risk to the ecosystem and human health. This study may provide a scientific basis for the prevention and management of sulfonamides pollution and the development of a healthy aquatic ecosystem in the Weihe River basin.

## 2. Materials and Methods

### 2.1. Study Area and Sampling Site Description

The Weihe River originates at the Niaoshu Mountain in Weiyuan County, Gansu Province, and flows through Baoji, Xianyang, Xi’an, and Weinan, and eventually converges into the Yellow River. Being the largest tributary of the Yellow River, the basin area of Weihe River (67,108 km^2^) accounts for 50% of the total area of the Yellow River basin in Shaanxi Province. The total length of the main stream of the Shaanxi section of the Weihe River (Linjia Village–Tongguan) is 379 km. There are many tributaries with asymmetric development on both sides within the study area, wherein more tributaries than the north bank characterize the south bank. Most tributaries on the North bank comprise relatively little water, while those on the south bank comprise clear water with a short source and rapid flow. The rivers other than the Heihe and Bahe Rivers are less than 100 km in length. This study distinguished the upper, middle, and lower reaches as Linjia Village–Tanyu River, Xianyang Dam–Xinfeng, and Shawangdu–Tongguan, respectively. The main stream and major tributaries of the Shaanxi section of the Weihe River are depicted in Figure 1 and Appendix A.

### 2.2. Methods

#### 2.2.1. Reagents and Chemicals

Reagents: Liquid chromatography–tandem mass spectrometer (LC–MS/MS, Agilent 1260 + 6460B QQQ with ESI ion source), nitrogen blowing apparatus (Biotage), and a solid-phase extraction apparatus (Dionex SPE-280).

Chemicals: Sulfadiazine (SDZ), sulfathiazole (STZ), sulfapyridine (SPZ), sulfamethazine (SM2), sulfamonomethoxine (SMM), sulfachloropyridazine (SCP), sulfamethoxazole (SMZ), sulfamethizole (SMI), sulfamethoxypyrimidine (SMR), sulfadimethoxine (SDM), sulfamethoxypyrimidine (SMP), and sulfaquinoxaline (SQZ) were purchased from Dr. Ehrenstorfer, Germany, and acetonitrile were purchased from Merck (Darmstadt, Germany). The formic acid used for mass spectrometry was purchased from J&K Scientific. The analytically pure hydrochloric acid and disodium EDTA (Na-2 EDTA-2H2O) were purchased from China National Pharmaceutical Group Corporation. Ultrapure water was purchased from a Milli-Q water purification system (Millipore, Burlington, MA, USA) and used for the experiments throughout the study.

#### 2.2.2. Sample Collection and Pre-Treatment

The water samples were collected during the wet water period in October 2020 and the flat water period in May 2021 from 31 sampling sites (W1~W31) with the same locations for both sampling periods. All sampling sites were set in the national control section, provincial control section, and key pollution sections of the Weihe River. The water samples in this study were snap samples since the water flow in the Weihe River basin was stable during the sampling period. All samples were collected according to the “water quality sampling program design technical requirements” (HJ495–2009), using a plexiglass water grab sampler from about 0.5 m below the surface. At each sampling location, three parallel control groups were set up with 2 mL of methanol to prevent microbial activity, and the water samples were stored at 4 °C. Water temperature (WT), electrical conductivity (EC), and total dissolved solids (TDS) were measured in the field using portable detectors. All samples were returned to the laboratory for processing and analysis within 24 h.

Pre-treatment method: Briefly, 1.0 L of water sample was filtered via a 0.45 μm pore-size glass fiber membrane, followed by the addition of 5 mL of 100 g·L^−1^ EDTA·2Na solution, and then 50% (*v*/*v*) phosphoric acid aqueous solution was used to adjust the pH value of the water sample to 3.0; afterward, 6 mL of methanol, 3 mL of ultrapure water, and 6 mL of sodium dihydrogen phosphate aqueous solution were added to the HLB cartridge in turn to activate the cartridge, and the flow rate of the cartridge solution was adjusted to 4 mL/min. After the sample was drained, 10 mL of ultrapure water was added to the sampling bottle, dried under vacuum for 30 min, and then eluted with 6 mL of methanol and 6 mL of 2% (*v*/*v*) ammonia methanol at a flow rate of 1 mL/min; the eluate was evaporated nearly dry at 40 °C and then redissolved with methanol to 1 mL for LC–MS/MS analysis.

#### 2.2.3. LC–MS/MS Analysis and Quality Control

The target contaminants in the samples were analyzed using liquid chromatography–tandem mass spectrometry (LC–MS/MS). The liquid chromatography (LC) parameters include mobile phase A of 0.01 mol/L formic acid, mobile phase B of water, 2:8 acetonitrile, 1.0 mL/min flow rate, and column temperature of 40 °C, 20 µL sample quantity, and C18 column of 4.6 × 150 mm, with 5 µm. The coefficient of determination (R^2^) of the standard curves of antibiotics was greater than 0.99 and was compared with the concentration of the tested antibiotic. Three water samples were collected at each sampling location for parallel analysis to ensure the reliability of the experimental result.

The test was performed according to the criterion that the parallel sample should be no less than 20% of the test sample. In this study, one parallel sample was inserted into five samples. The recovery rate of the test was between 85% and 120%.

#### 2.2.4. Ecological Risk Assessment

The potential ecological risk induced by the exposure to sulfonamides in the Shaanxi section of the Weihe River was evaluated using the method adopted in the European Technical Guidance Document for environmental risks of pollutants (TGD) [21]. Using RQs, the ecological risk of sulfonamides in the water environment was calculated using the following relation:(1)$$\mathrm{RQs}=\frac{\mathrm{MEC}}{\mathrm{PNEC}}$$
where RQs represent the risk quotient; MEC represents the measured environmental concentration (ng/L), and PNEC represents the predicted no-effect concentrations (ng/L).

The PNEC is calculated as follows:(2)$$\mathrm{PNEC}=\frac{\left({\mathrm{EC}}_{50}\right)\mathrm{or}\left({\mathrm{LC}}_{50}\right)}{\mathrm{AF}}$$
where EC_50_ or LC_50_ represents acute or chronic toxicity data; AF represents the assessment factor which is 1000 and 100 for acute and chronic toxicities, respectively. The PENC values and EC_50_ or LC_50_ (Table 1) were derived from the EPA ECOTOX toxicity database (https://cfpub.epa.gov/ecotox/, accessed on 5 April 2022) and toxicological data from the published literature [22].

In accordance with the RQ_S_ classification method for determining the ecological risk level [23], the RQ_S_ was divided into four risk levels, which are RQ_S_ < 0.01, 0.01 ≤ RQ_S_ < 0.1, 0.1 ≤ RQ_S_ < 1, and RQ_S_ ≥ 1, representing no significant risk, low risk, medium risk, and high risk, respectively.

#### 2.2.5. Health Risk Assessment

The RQ_H_ was calculated based on the ADI of humans and the RQs calculation model [24], and the RQ_H_ of antibiotics to human health in the Shaanxi section of the Weihe River was calculated to assess the health risk of SAs, using the following relation:(3)$${\mathrm{RQ}}_{H}=\frac{\mathrm{MEC}}{\mathrm{DWEL}}$$
where RQ_H_ represents the health risk entropy value for a single antibiotic, MEC represents the measured concentration of antibiotics (µg/L), and the DWEL represents the drinking water equivalent value (µg/L).

The DWEL was calculated using the following relation:(4)$$\mathrm{DWEL}=\frac{\mathrm{ADI}\times \mathrm{BW}\times \mathrm{HQ}}{\mathrm{DWI}\times \mathrm{AB}\times \mathrm{FOE}}$$
where ADI represents the allowable daily intake µg/(kg·d); BW represents the body weight (kg); HQ represents the highest risk, which is assumed as 1; DWI is the daily water intake (L/d); AB represents gastrointestinal absorption rate, which is assumed as 1; and FOE is the frequency of exposure (350 d/a), which is 0.96. The BW, DWI, and antibiotic ADI values for various age groups are listed in Table 2 and Table 3.

The ADI values for SMI and SMP were not found; thus, only the existing data are listed in Table 3. Further, the RQ_H_ was divided into four risk levels to human health based on various reports [27]: RQ_H_ ≥ 1, 0.1 ≤ RQ_H_ < 1, 0.01 ≤ RQ_H_ < 0.1, and RQ_H_ < 0.01 representing high risk, medium risk, for low risk, and no significant risk.

## 3. Results and Discussion

### 3.1. Concentrations of Sulfonamides in the Shaanxi Section of the Weihe River

The concentration and detection rates of sulfonamides in the Shaanxi section of the Weihe River are shown in Table 4, which illustrates the presence of five SAs (SDZ, SM2, SMM, SMZ, SQZ) during the wet water period. The detection rate of SMZ reached 90.30% at all the sampling sites, with a maximum concentration of 34.256 ng/L, which was much higher than the other four antibiotics. Similarly, the frequency rates of SQZ and SMM were also relatively high, which were 54.8% and 51.6%, respectively, followed by SDZ and SM2 at 12.9%. Additionally, five types of SAs (SPZ, SM2, SMM, SMZ, and SQZ) were detected during the flat water period, with lower concentrations than those observed in the previous period. The highest detection rates of 13.3% were observed for SMM and SQZ, while the detection rates of SPZ, SM2, and SMZ were 10%, 8.57%, and 8.57%, respectively. Significant differences were noted in the occurrence of the SAs, as revealed in Table 4, and this can be attributed to the use of antibiotics and their varying environmental behavior throughout the time periods.

SMZ was the major pollutant during the wet water period, with a significantly higher concentration level and detection rate than the other four sulfonamides, as shown in Figure 2. This is due to the widespread use of SMZ for the prevention and treatment of influenza or respiratory infections, often breaking out during spring and autumn [28], and also the common usage of SMZ as an antimicrobial drug for human and veterinary due to its wide antimicrobial spectrum, strong antimicrobial properties, high quality, and low cost. As a result, SMZ is used in large quantities and more often than other antibiotics. Moreover, antibiotic concentration levels are also related to their physicochemical properties, such as poor degradation capacity in the aquatic environment and the longer migrating capacity, which enables SMZ detection in any sampling sites at varying degrees. Conversely, SPZ and SQZ were the two dominant antibiotics in water during the flat water period. All the remaining SAs, except for SDZ, were present at low levels, indicating a relatively low residual concentration of the five antibiotics even with different degrees of usage. There were seasonal fluctuations in pollutant levels due to variations in antibiotic use and environmental conditions.

Currently, the consumption of antibiotics in various parts of China varies significantly, with greater quantities consumed in coastal areas than inland areas and southern regions than northern regions, which is also correlated with the regional economic development. A study conducted on the use of antibiotics in different regions of China in 2013 by Zhang et al. [29] shows that, among the seven regions in China, the maximum consumption of SAs is in the east, and the minimum consumption is in the northwest, as shown in Figure 3. The Weihe River flows through the eastern Gansu and central Shaanxi Provinces, both of which are located in northwestern China. In this research, the pollution and consumption of SAs correspond to each other.

Antibiotic pollution in different regions is influenced by a number of factors, including population, economic pillar industries, and antibiotic consumption level. It is also related to the spatial variations in the climatic conditions, geographical environment, and the physicochemical properties of antibiotics. In this context, the pollution levels of SAs were compared in the Guangzhou section of the Pearl River, Nanjing section of the Yangtze River, Haihe River, Liaohe River, Songhua River, and Shaanxi section of the Weihe River, which is located in the southern, eastern, central, northern regions of China. The concentration of SAs in the Shaanxi section of the Weihe River was generally lower than those in the domestic and foreign rivers, as seen in Table 5. However, the SMZ was higher than that in certain rivers due to its presumably high abundance in the Weihe River during the earlier period. Moreover, SMZ exhibited a moderate concentration in the Weihe River basin, similar to the Ebro River, and was significantly higher than in the Nanjing section of the Yangtze River, Hong Kong river, and Bangladesh River, indicating serious overconsumption of SMZ in this study area [30]. Except for undetected antibiotics, the overall concentrations of SDZ and SMZ in the Bangladesh River were lower than those detected in this study.

Among these typical rivers above, the Guangzhou section of the Pearl River flows through the economically developed region of the Pearl River Delta and has a very substantial discharge of domestic sewage, breeding wastewater, and medical wastewater; as a result, antibiotics have been identified in this aquatic environment to varying degrees; the water quality in the Yangtze River, as the most important industrial water and drinking water source in China, has a direct influence on the surrounding residents and industrial production. The Nanjing section of the Yangtze River is situated in the lower reaches of the Yangtze River and is the most economically developed region in China. The Haihe River is the most extensive water system in northern China, covering most of the areas, including Beijing, Tianjin, Hebei, and Shandong, with a large population and abundant natural resources [37], even though the river runoff is relatively low; both Liaohe and Songhua Rivers are located in the northeast of China, with a well-developed farming industry around the basin. SAs such as widely used antibiotics for humans and animals are detected at significantly high concentrations. Although the Yellow River is the second largest river in China, there are few reports on the presence of antibiotics in it [38]. Antibiotic concentrations in Yellow River mainstem and tributary streams varied from 3 to 56 ng/L, with low detection rates, compared with those of other rivers [39]. In conclusion, the economically developed areas and areas with developed farming industries in China have extensively been using and discharging SAs, resulting in serious antibiotic pollution in these watersheds. Therefore, the high level of antibiotics in the area can be attributed to the high input and high consumption of antibiotics. Moreover, the average annual precipitation in China generally decreases from southeast to northwest due to the influence of the monsoon, concentrating mainly during summer. This differential rainfall leads to significant differences in the surface water runoff in different areas, which favors the entry of antibiotics into the surface water causing varying pollution levels in different watersheds.

### 3.2. Spatial and Temporal Distribution of Sulfonamides in the Shaanxi Section of the Weihe River

#### 3.2.1. Temporal Distribution of Sulfonamides in the Shaanxi Section of the Weihe River

From Figure 4, it is evident that the concentration levels and detection rates of SAs in the Shaanxi section of the Weihe River during the two periods differed significantly, with higher rates during the wet water period than those found during the flat water period. The concentration during the wet water period typically ranged from ND to 34.256 ng/L, with detection rates ranging from 12.9% to 90.30%. Similarly, during the flat water period, the concentration ranged from ND to 2.113 ng/L, with detection rates ranging from 8.57% to 13.3%. Different antibiotics often exhibit different seasonal effects [40], which can be attributed to the increasing frequency of livestock and aquaculture activities and SA consumption in the surrounding medical industry during this period. In addition, the Weihe River basin receives the majority of its precipitation from June to October, which is also the season for frequent disease, livestock growth, and agricultural planting, which necessitates the use of large quantities of manure and organic fertilizers containing antibiotics [41]. The SAs are easily transported to the river through surface runoff [42] during the rainy season due to their low adsorption capacity in soil or manure. Moreover, when the temperature decreases, the light intensity, hydrodynamic conditions, and microbial degradation ability also deteriorate, reducing the antibiotic degradation rate in the aquatic environment. The low temperature also affects the removal effect of these compounds in wastewater treatment plants [43], resulting in a considerable amount of antibiotics being released into the river during the wet water season that has not entirely been removed. The concentration levels and detection rates of SAs during the flat water period decreased significantly due to the use of antibiotics and the enhanced biodegradation in response to the gradual rise in temperature, which catalyzes the dilution effect, and photolysis of SAs in the water environment [44]. This promoted the transformation and decomposition of antibiotics, which subsequently resulted in relatively low pollution levels during this period.

Several scholars have investigated the antibiotic contamination in the Weihe River basin in recent years. The contamination levels of SAs in the Weihe River during 2016–2018 were significantly higher than those observed in this study [45,46,47]. However, during the flat water period, in this study, SMZ was detected at only two sampling sites where the concentration was just 1.3 ng/L. The pollution has been decreasing significantly in the Shaanxi section of the Weihe River, which is evident from the concentration level of SAs from 2016 to 2021, and this can be related to the management of antibiotics in China. The Ministry of Agriculture and Rural Affairs continuously implemented strategies to reduce veterinary antimicrobial drugs on various farms across the country from 2018 to 2020. Additionally, the General Office of the National Health and Wellness Commission issued a notice on the continuous improvement of the management of the clinical application of antimicrobial drugs in 2020, which requires in-depth implementation of the “Opinions on Strengthening Pharmacy Management in Medical Institutions” to promote rational drug use and the National Action Plan to curb bacterial drug resistance (2016–2020) to improve the rational use of antimicrobial drugs continuously. This resulted in a significant variation in the level of SAs contamination in this study, compared with those observed in previous studies.

#### 3.2.2. Spatial Distribution Characteristics of Sulfonamides in the Shaanxi Section of the Weihe River

As part of this study, 31 sampling sites (3 sampling sites in the north tributary, 16 sampling sites in the main stream, and 12 sampling sites in the south tributary) during the wet water period and 30 sampling sites (3 sampling sites in the north tributary, 15 sampling sites in the main stream, and 12 sampling sites in the south tributary) during the flat water period were set up, and SA distribution across the sampling sites is shown in Figure 5.

During the wet water period, the majority of the antibiotic types were detected at W22 and W28, with a total of 4 SA types in the 31 sampling sites, covering the Geng Town and Shi Village with concentrations, ranging from 0.358 to 18.338 ng/L and from 0.23 to 17.95 ng/L, respectively, as shown in Figure 5. More than 1 type of SAs was detected in the remaining 27 sampling sites, but only 2 sampling sites had no SAs, indicating the prevalence of SAs throughout the upper, middle, and lower reaches during the wet water period. During the flat water period, only 2 or more SAs were detected at W10 and W19, which were located at Yangling Dam and Tianjiang Rendu, respectively, and 1 antibiotic was detected in the remaining 10 sampling sites, while no SAs was detected in any of the 18 sampling sites, suggesting the significant contribution of SAs in the middle reaches during this period.

The distribution of SAs in the main and tributary streams of the Weihe River during two periods is shown in Figure 6. The order of distribution of SAs from the highest to the lowest during the wet water period was main stream (ND–35.296 ng/L) > the south tributary (3.718–34.354 ng/L) > the north tributary (5.476–9.302 ng/L), and that during the flat water period was main stream (ND–3 ng/L) > the north tributary (ND–2.095 ng/L) > the south tributary (ND–1.3 ng/L). Baoji, on the upper reaches of the Shaanxi part of the Weihe River, is the location of sampling sites W1–W8. It is the second-largest industrial city in Shaanxi Province, and the predominant industries in the city include machinery, chemicals, food processing, etc. These industries use water and release a substantial volume of effluents into the Weihe River [48]. The highest concentration of SAs in the upper main stream was observed at W1 (Linjia village) due to the vast population and family farming base in the area, where villagers and livestock cannot survive without antimicrobials. Sampling sites W9–W24 are located in the middle reaches and may be subdivided into three sections based on population density, agricultural industry, and factory density: The upper area in Xianyang, the midstream area in Xi’an, and the downstream area in Gaoling and Lintong. The population density progressively rises downstream from the source, and the sampling points W9 and W10 were located above and below the Yangling dam, respectively. The construction of the dam exerts an intercepting effect on the river, which slows the flow of water at this location and limits the dilution rate of antibiotics, resulting in a greater concentration of SAs downstream than upstream. The W19–W22 were located near the Bahe River, where many poultry and livestock breeding bases exist in the surrounding area, and the outfall of Xi’an City No. 5 Wastewater Treatment Plant is also located there; the discharge of pharmaceutical wastewater, breeding wastewater, and domestic wastewater resulted in increased antibiotic concentrations. The sampling sites W25–W31 were located in the lower reaches, with even distribution of the main stream and tributaries, and the sampling site W27 was located in the Chishuihe River, a tributary on the south bank; since the upstream of this site flows through two outfalls (the Weinan Municipal Sewage and the Weinan West Alkali Drainage Canal), the effluent from the sewage treatment plant may be the primary cause of the elevated SAs at this site [49]. When domestic sewage or agricultural wastewater is discharged, or when the river runoff and river velocity are low in the tributaries, SAs converge into the main stream and continue to accumulate, resulting in high levels of SAs pollution in the main stream. During the flat water periods, SAs levels in the main stream and its tributaries were generally low, and the overall antibiotic concentration in each sample was comparable. The sampling site W19 was located at Tianjiang Rendu, which is downstream of the Zaohe River and the Caoyun nullah, wherein the former river is commonly known as the “sewage river” of Xi’an. Domestic sewage and industrial wastewater are often discharged into the tributaries and eventually into the main stream, resulting in a higher detection level of SAs at the Tianjiang Rendu section.

In conclusion, the variations in the concentrations of SAs in the Shaanxi section of the Weihe River may be related to pollution sources, population density, local economic development levels, and agricultural farming on both sides of the basin. In addition, relatively severe pollution of SAs in main stream of the Weihe River is also related to the tributaries and the outfall canals, which are both the recharge sources for the main stream and the input sources for the downstream pollutants. The very high discharge of antibiotics by various industrial enterprises, pharmaceutical companies, and outfalls in the tributaries on both banks may increase the antibiotic residues in the main stream, resulting in a higher pollution level of SAs in the main stream of the Weihe River than in the tributaries.

### 3.3. Ecological Risk Assessment of Sulfonamides in the Shaanxi Section of the Weihe River

The RQ values of SAs varied in the wet water period, as illustrated in Figure 7. In order to comprehend the risk levels of antibiotics in the Shaanxi section of the Weihe River, the risk assessment results of this study were assessed using the toxicity data and PNEC values of various sulfonamides to aquatic organisms such as fish, algae, water flea, etc. The RQ values of SM2, SMM, and SQZ for respective sensitive species were less than 0.01 at each of the 31 sampling locations during the wet water period, indicating that the ecological risk caused by these three SAs was not significant. On the other hand, SDZ exhibited low risk (0.01 ≤ RQ_s_ < 0.1) to water flea and *Microcystis* sp. only at sampling site W1; SMZ showed low risk (0.01 ≤ RQ_s_ < 0.1) to *Cyclotella* sp. in sampling sites W1, W9, W10, and W27, but no significant ecological risk to other aquatic organisms in other sampling locations. As a result, the ecological risk caused by SDZ and SMZ should be widely acknowledged. Since none of the five antibiotics detected during the flat water period posed a significant ecological risk (RQ_s_ < 0.01) to any of the aquatic organisms in the watershed, Figure 7 only depicts the ecological risk of SAs to aquatic organisms during the wet water period. Furthermore, SDZ and SMZ are the primary sources of ecological risk. These two sulfonamides are also widely used for medicine and agriculture, which may have toxic effects on aquatic organisms, promote the long-term development of bacterial resistance in the water environment [50] and may potentially impact the original ecosystem.

This study used the most sensitive species corresponding to SAs as the benchmark, calculated the MEC based on the highest observed concentration, and analyzed the changes in RQ values of sulfonamides over the last five years. As shown in Figure 8, the RQ_S_ of the six SAs for various aquatic organisms exhibited a declining trend from 2016 to 2021 and generally showed an order of wet water period > flat water period. In 2016, SQZ was greater during the flat water period than that during the wet water period, which may have been caused by the presence of aquatic organisms along the watershed. In conclusion, the ecological risk has changed since the start of the antibiotic reduction action in the Shaanxi section of the Weihe River, also indicating that individual antibiotics are still abused. Therefore, more attention should continue to be paid to the use of SAs in the Weihe River in order to prevent more severe ecological threats to aquatic organisms.

### 3.4. Health Risk Assessment of Sulfonamides in the Shaanxi Section of the Weihe River

Figure 9 compares the health risks of sulfonamides on human health throughout different periods from 2016 to 2021, where (c) and (d) represent the health risks of six SAs to various age groups during the wet water period in 2020 and the flat water period in 2021, respectively. The RQ_H_ values for various age groups varied from 3.17 × 10^−7^ to 9.49 × 10^−6^, showing that there was no major health risk (RQ_H_ < 0.01) for SAs in the Shaanxi section of the Weihe River. Nonetheless, the six SAs revealed distinct health risks for various age groups, with RQ_H_ values greater for adults than children and males than females in the same age group. Compared with the values of 2016, the RQ_H_ values observed in this study decreased by one order of magnitude, as seen in Figure 9. Long-term exposure to emerging contaminants at low concentrations can be harmful to human health [51], and it can also lead to the development of antibiotic-resistant bacteria (ARB) and antibiotic resistance genes (ARGs), which can increase the risk to human health even at low concentrations through the food chain and food web [52].

As an emerging contaminant, antibiotic resistance poses a growing threat to global public health. Emerging contaminants are “emerging” because they are biotoxic, persistent in the environment, and bioaccumulative. Even at low concentrations in the environment, they may cause significant risks to the environment and health. Antibiotics are not currently incorporated into environmental management, or existing management measures are inadequate, necessitating action from all government agencies and society [53]. The WHO has identified antimicrobial resistance as a threat to public health in the 21st century [54]. In recent years, China has repeatedly emphasized the governance of emerging contaminants, moving from “special research on the governance of emerging contaminants” to “pay attention to the governance of emerging contaminants” and, finally, “strengthen the governance of emerging contaminants.” The demands for new pollutant governance have increasingly developed, rising in severity, and the importance of the task has been emphasized. The findings of this study might be relevant for future antibiotic pollution control measures in the Weihe River basin. This study did not explore the risks posed by antibiotic pollutant interactions and resistance genes, which solely looked at the impacts of six SAs on various aquatic organisms and human health. Antibiotic compound pollution poses additional ecological and health risks, which should be addressed in future studies.

## 4. Conclusions


(1)In the Shaanxi section of the Weihe River, SAs are abundant. In this basin, six SAs (SDZ, SPZ, SM2, SMM, SMZ, and SQZ) were detected, with five SAs being detected in each of the wet and flat water periods, with detection rates varying from 12.9% to 90.30% and from 8.57% to 13.3%, respectively. In comparison to other domestic and foreign rivers, SA pollution in the Shaanxi section of the Weihe River was rather low, but the risks caused by individual antibiotic usage should still be considered.(2)In terms of temporal distribution, SAs exhibited seasonal changes associated with rainfall, light intensity, ambient temperature, and biological and microbial activity, with wet water period (ND–34.256 ng/L) > flat water period (ND–2.113 ng/L). In terms of spatial distribution, it was revealed that the order was the main stream (ND–35.296 ng/L) > the south tributary (3.718–34.354 ng/L) > the north tributary (5.476–9.302 ng/L) during the wet water period, and the main stream (ND–3 ng/L) > the north tributary (ND–2.095 ng/L) > the south tributary (ND–1.3 ng/L) during the flat water period, which is directly associated with the enterprises on both sides and the degree of population density, economic development, and agricultural farming.(3)Except for SDZ and SMZ, which exhibited low risk (0.01 ≤ RQS < 0.1) to water flea and Microcystis sp. in some sampling sites during the wet water period, no other antibiotics showed a substantial ecological risk (RQS < 0.01) to their sensitive species. The risks of six SAs to aquatic organisms have decreased in the last five years, with the order of wet water period > flat water period, which is intrinsically connected to the development of emerging contaminant management in recent years.(4)In the Shaanxi section of the Weihe River, there was no significant health risk (RQH < 0.01) of SAs in different age groups, and the risks for adults were greater than for children, and in the same age group, males were at higher risk than females. The RQH values in this study declined by an order of one magnitude, compared with those of 2016. However, the risks from antibiotic contaminant interactions and resistance genes were not explored in this study, which solely looked at the effects of the six SAs on diverse aquatic organisms and human health.


## Figures and Tables

**Figure 1 ijerph-19-08607-f001:**
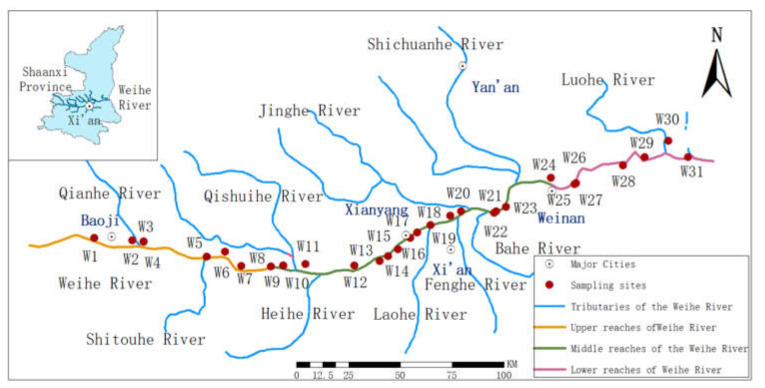
Distribution of sampling sites in Shaanxi section of the Weihe River.

**Figure 2 ijerph-19-08607-f002:**
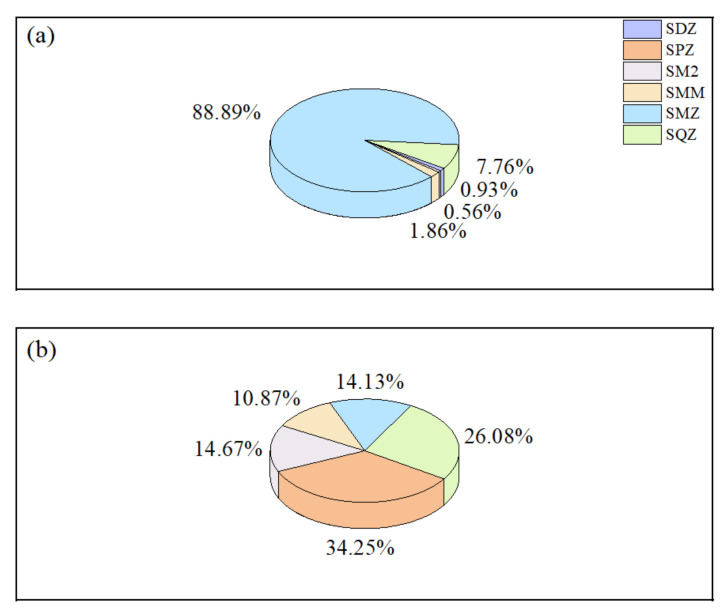
The proportion of sulfonamides: (**a**) the wet water period; (**b**) the flat water period.

**Figure 3 ijerph-19-08607-f003:**
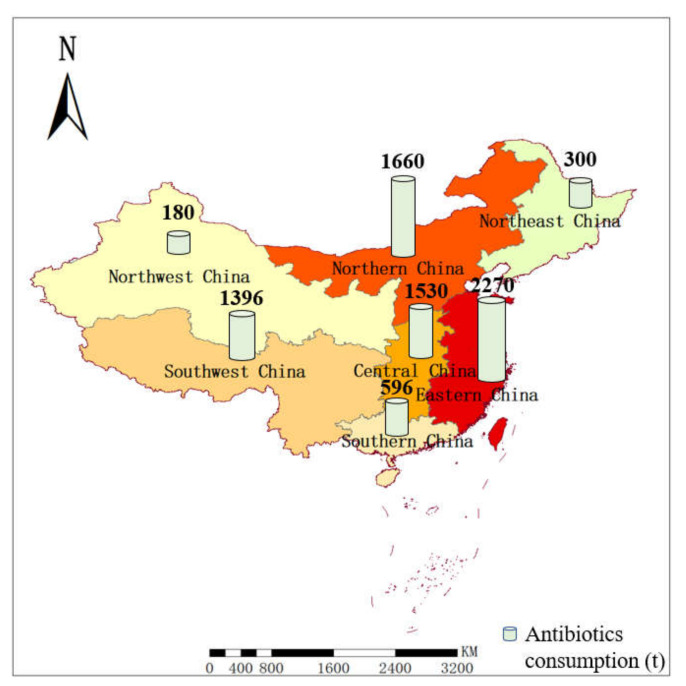
Consumption of antibiotics in different regions of China.

**Figure 4 ijerph-19-08607-f004:**
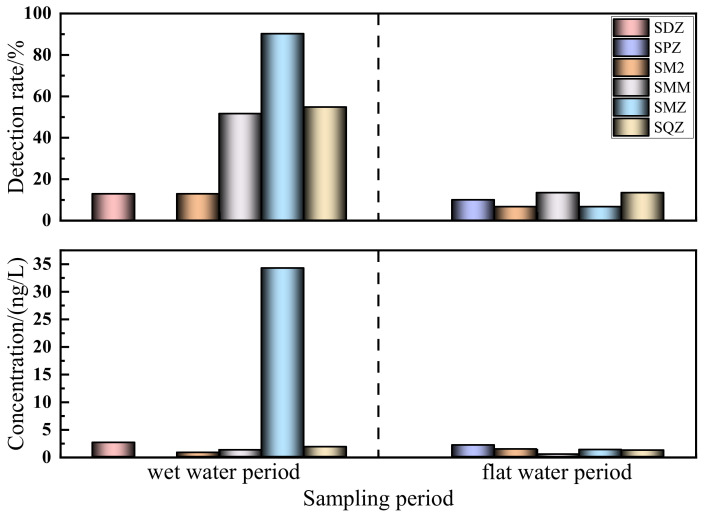
Seasonal distribution of SAs detection rate and concentration in the Shaanxi section of the Weihe River.

**Figure 5 ijerph-19-08607-f005:**
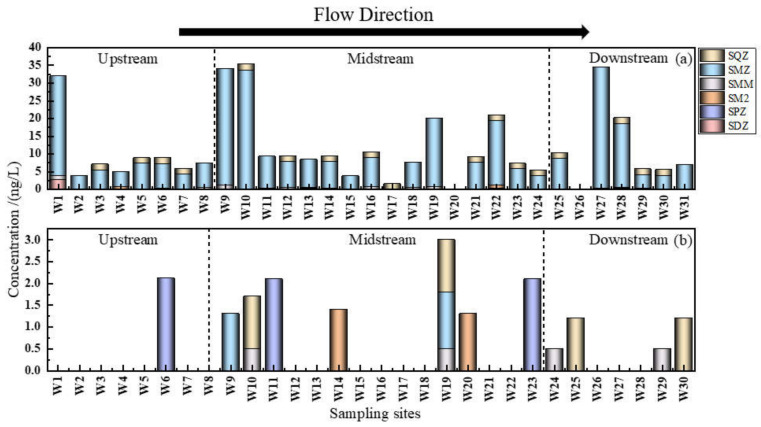
Distribution of SAs in the Shaanxi section of the Weihe River: (**a**) the wet water period; (**b**) the flat water period.

**Figure 6 ijerph-19-08607-f006:**
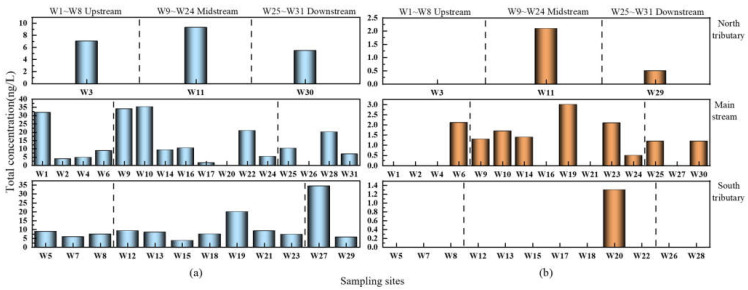
Distribution of SAs in the main steams and tributaries of the Weihe River: (**a**) the wet water period; (**b**) the flat water period.

**Figure 7 ijerph-19-08607-f007:**
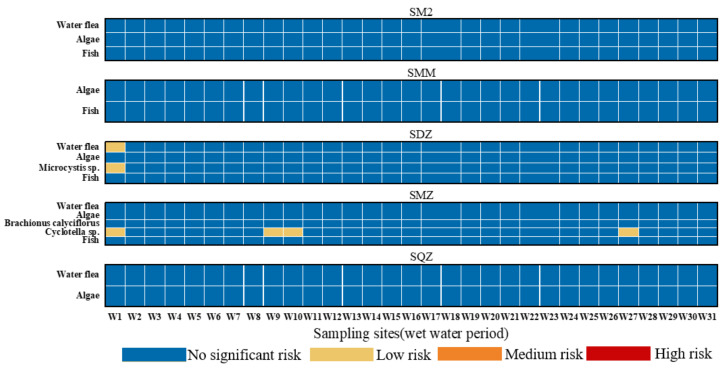
The RQ_s_ of SAs in Shaanxi section of the Weihe River.

**Figure 8 ijerph-19-08607-f008:**
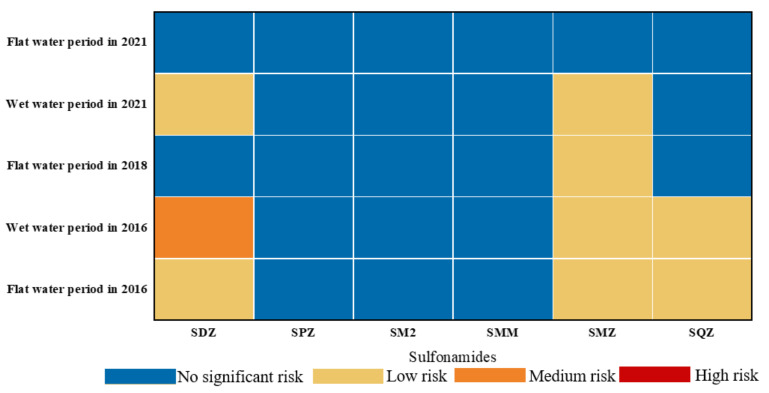
Comparison of the RQ_s_ of sulfonamides in Shaanxi section of the Weihe River from 2016 to 2021.

**Figure 9 ijerph-19-08607-f009:**
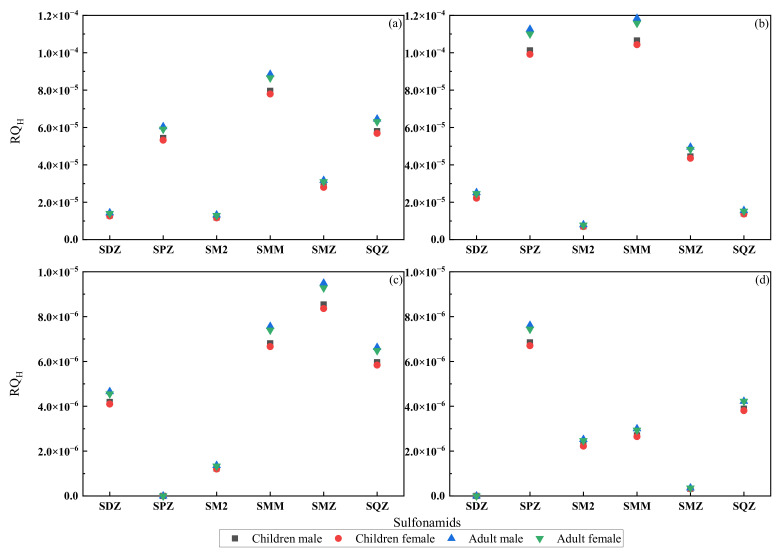
Comparison of the RQH in Shaanxi section of the Weihe River from 2016 to 2021: (**a**) the flat water period in 2016; (**b**) the wet water period in 2016; (**c**) the wet water period in 2020; (**d**) the flat water period in 2021.

**Table 1 ijerph-19-08607-t001:** Toxicity data and PNEC values for sensitive species to antibiotics.

Sulfonamides	Corresponding Sensitive Species	EC_50_/(mg/L)	Toxicity Type	AF	PNEC/(ng/L)
SDZ	Fish	890	Acute toxicity	1000	890,000
	*Microcystis* sp.	0.135			135
	Algae	0.52			520
	Water flea	0.21			210
SPZ	Lemna minor	0.46	Acute toxicity	1000	460
	*Chlorella vulgaris*	5.28			5280
SM2	Fish	517	Acute toxicity	1000	517,000
	Algae	38			38,000
	Water flea	4			4000
SMM	Fish	450	Acute toxicity	1000	450,000
	Algae	8.56			8560
SMZ	Fish	890	Acute toxicity	1000	89,000
	*Cyclotella* sp.	2.4			2400
	*Brachionus calyciflorus*	26.27			26,270
	Algae	51			51,000
	Water flea	4.5			4500
SQZ	Algae	20	Acute toxicity	50	400
	Water flea	84.46		1000	84,460
SCP	*Chlorella vulgaris*	32.25	Acute toxicity	1000	32,250

**Table 2 ijerph-19-08607-t002:** BW and DWI values for adults and children.

Research Subjects	Gender	BW/kg	DWI/L·d
Children	Male	24	0.81
	Female	23	0.76
Adults	Male	66.1	2.48
	Female	57.8	2.12

**Table 3 ijerph-19-08607-t003:** ADI values of different sulfonamides.

Sulfonamides	ADI/(μg·kg^−1^·d^−1^)	References
SDZ	20	[24]
SPZ	10	
SM2	20	
SMM	6	[25]
SMZ	130	[24]
SQZ	10	[25]
SDM	10	
SCP	50	[26]
SMR	50	
STZ	50	

**Table 4 ijerph-19-08607-t004:** Concentration levels of sulfonamides in the Shaanxi section of the Weihe River.

Sulfonamides	Wet Water Period				Flat Water Period			
	Min/(ng/L)	Max/(ng/L)	Mean/(ng/L)	Detectionrates/%	Min/(ng/L)	Max/(ng/L)	Mean/(ng/L)	Detection Rates/%
SDZ	ND	2.584	0.106	12.90	ND	ND	ND	0
SPZ	ND	ND	ND	0	ND	2.113	0.210	10.00
SM2	ND	0.756	0.064	12.90	ND	1.4	0.090	6.70
SMM	ND	1.260	0.212	51.60	ND	0.500	0.067	13.30
SMZ	ND	34.256	10.126	90.30	ND	1.3	0.087	6.70
SQZ	ND	1.84	0.884	54.80	ND	1.2	0.160	13.30

ND, not detected.

**Table 5 ijerph-19-08607-t005:** The concentrations of sulfonamides in domestic and foreign rivers (ng/L).

Name	Concentration/(ng/L)							References
	Item	SDZ	SPZ	SM2	SMM	SMZ	SQZ	
Shaanxi section of the Weihe River	Max	2.584	ND	0.756	1.26	34.256	1.84	-
	Min	ND	ND	ND	ND	ND	ND	
Guangzhou section of the Pearl River	Max	13.7	10.4	256	56.8	210	3.02	[12]
	Min	3.5	ND	8.2	9.14	2.66	ND	
Nanjing section of the Yangtze River	Max	6.59	1.01	-	ND	6.76	-	[13]
	Min	2.52	0.36	-	ND	8.98	-	
Haihe River	Max	270	-	940	-	660	-	[14]
	Min	ND	-	ND	-	ND	-	
Liaohe River	Max	-	0.96	15.91	-	670.27	14.59	[15]
	Min	-	ND	ND	-	ND	ND	
Songhua River	Max	505	85.0	16.1	ND	940	-	[31]
	Min	0.86	ND	ND	ND	ND	-	
Hongkong River	Max	14.8	3.2	580.4	-	3.1	-	[32]
	Min	1	0.9	16.5	-	1.1	-	
Bangladesh River	Max	0.58	-	11.35	-	7.24	-	[33]
	Min	0.03	-	0.02	-	0.03	-	
Mekong River	Max	-	-	328	-	174	-	[34]
	Min	-	-	15	-	20	-	
Ebro River	Max	6.4	-	ND	-	35.6	-	[35]
	Min	1.3	-	ND	-	1.88	-	
Youngshan River	Max	20	-	20	-	110	-	[36]

ND, not detected.

## Data Availability

Not applicable.

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
