# Peer review of "Spatial and Temporal Distribution Characteristics and Potential Risks of Sulfonamides in the Shaanxi Section of the Weihe River"

_ijerph, 2022, doi:10.3390/ijerph19148607_

Round 1
Reviewer 1 Report
This manuscript describes the occurrence and environmental risk of Sulfonamides in the Shaanxi Section of the Weihe river. The work's theme is relevant and up-to-date, but some flaws should be addressed before it can be published. Therefore, in my opinion, the paper may only be published after major revision.
My opinions and comments about this manuscript are as follows:
1. Abstract: Do not use abbreviators in the abstract. The first mention of an abbreviation must be accompanied by the full name
2. Table 1 should be moved to Material Supplementary.
3. 2. 2.2. Sample collection: you do not describe well the collection procedure which is very important for the representativeness of the sample. Was the sample mixed with other sub samples? The sample collection should be presented in more detail.
4. Does the analysis of the samples need any previous treatment, or were the samples analysed directly on the HPLC system? The process should be explained
5. the equation 1-4 are missing
6. Quantification: The validation procedure should be show. Important validation parameters (as inter-day accuracy and inter-day precision, matrix effects) were not established. What about linearity range of the analytical method?
7. The figure 2 and table 5 show the same information. keep figure 2 and move table 5 to supplementary material.
Reviewer 2 Report
In this study, researchers focused on the concentrations of six contaminants analyzed from river water in two seasons of the year, Oct 2020 and May 2021. The manuscript presents a complex analysis of the Shaanxi Section, but the data presentation is not very clear and needs to be improved.
Nevertheless, I have some comments that made my decision that I recommend accepting this paper after a major review.
The comments are indicated in the attached file.

Round 2
Reviewer 1 Report
The paper has improved and the authors have adequately responded to the questions presented
Reviewer 2 Report
Dear authors,
I believe that overall, the paper deserves to be published in present form.